# Sympathetic Arousal Detection in Horses Using Electrodermal Activity

**DOI:** 10.3390/ani13020229

**Published:** 2023-01-07

**Authors:** Kia Golzari, Youngsun Kong, Sarah A. Reed, Hugo F. Posada-Quintero

**Affiliations:** 1Department of Biomedical Engineering, University of Connecticut, Storrs, CT 06269, USA; 2Department of Animal Science, University of Connecticut, Storrs, CT 06269, USA

**Keywords:** sympathetic arousal, horses, electrodermal activity

## Abstract

**Simple Summary:**

Monitoring stress in horses continuously can help us improve their quality of life. Currently, the detection of stress in horses relies on the observation of their behavior. This cannot be implemented continuously, as it is impossible to have someone looking at the horse all the time. One way to monitor stress in a continuous and automatic way is collecting the heart rate. The heart rate detects the arousal caused by stress, but it can be affected by physical activity and other emotions. Another way to assess the arousal caused by stress is measuring the changes in the conductance of the skin, called electrodermal activity (EDA). EDA can detect stress and pain in humans better than the heart rate. We have collected EDA in horses for the first time and evaluated its capability to detect arousal. We caused arousal in the horses using two tests. First, we collected EDA while the horses were being fed, which causes continuous arousal. Second, we used an umbrella to cause a startle and short-lasting arousal. EDA was sensitive to both tests. This shows that EDA is sensitive to arousal in horses and can be potentially used to detect stress and pain continuously.

**Abstract:**

The continuous monitoring of stress, pain, and discomfort is key to providing a good quality of life for horses. The available tools based on observation are subjective and do not allow continuous monitoring. Given the link between emotions and sympathetic autonomic arousal, heart rate and heart rate variability are widely used for the non-invasive assessment of stress and pain in humans and horses. However, recent advances in pain and stress monitoring are increasingly using electrodermal activity (EDA), as it is a more sensitive and specific measure of sympathetic arousal than heart rate variability. In this study, for the first time, we have collected EDA signals from horses and tested the feasibility of the technique for the assessment of sympathetic arousal. Fifteen horses (six geldings, nine mares, aged 13.11 ± 5.4 years) underwent a long-lasting stimulus (Feeding test) and a short-lasting stimulus (umbrella Startle test) to elicit sympathetic arousal. The protocol was approved by the University of Connecticut. We found that EDA was sensitive to both stimuli. Our results show that EDA can capture sympathetic activation in horses and is a promising tool for non-invasive continuous monitoring of stress, pain, and discomfort in horses.

## 1. Introduction

The sole absence of pain and discomfort is an indicator of animal welfare [1]. For that reason, continuous monitoring of those negative feelings is key to preventing stress and providing a good quality of life for horses. Several subjective techniques exist for welfare assessment, using protocols concentrating on feeding, health, and behavioral information [2,3,4]. These subjective tools do not allow continuous monitoring. Traditionally, mean heart rate (HR) and heart rate variability (HRV) have been used to assess autonomic arousal in horses [5,6]. HRV is also used in humans to assess autonomic arousal, but recently, objective pain and stress monitoring has been implemented using electrodermal activity (EDA), as EDA is a more sensitive and specific measure of sympathetic arousal than HRV [7]. The EDA signal is also referred to in the literature as galvanic skin response (GSR) and represents the conductance of the skin, which is modulated by sweat secretion (electrolyte solution) and has been gaining attention recently as a sensitive marker of sympathetic arousal because sweat glands are innervated solely by the sympathetic nervous system. However, to the best of our knowledge, no studies have evaluated the feasibility of EDA signal collection in horses.

HRV is a traditional measure that quantifies the fluctuation in time between heartbeats. HRV is controlled by the autonomic nervous system (ANS) to maintain cardiovascular homeostasis and normal blood pressure. HRV can be used to assess the sympathetic and parasympathetic branches of the ANS [8]. Because the higher frequencies (HFs) of HRV are solely affected by the parasympathetic branch, and both sympathetic and parasympathetic branches influence low-frequency (LF) components, HF and LF are used as markers of parasympathetic and sympathetic autonomic control, respectively, and the LF/HF ratio is used to assess the sympathetic and parasympathetic balance.

Several studies have used HRV for animals, including cattle, dogs, small ruminants, pigs, poultry, rodents, and horses [2,3,9,10,11,12,13,14,15,16,17,18,19,20,21]. For example, a visual Startle test (abrupt opening of an umbrella near the horse) was used as an acute stressor to activate autonomic reaction, and significant changes in the frequency indices of HRV were found [22]. Other studies evaluated stress and temperamental traits in horses [5,23,24,25]. Physical effort was reported to affect HRV [25,26], reducing the specificity of the tool for stress assessment. Furthermore, the use of LF as a measure of the sympathetic activity of the ANS is controversial since the low-frequency band also contains parasympathetic dynamics.

For its part, EDA is solely dependent on the sympathetic branch of the ANS because sweat glands are only innervated by this branch, which has made EDA an ideal target for detecting emotions [27]. Therefore, EDA can be used to quantitatively assess the sympathetic autonomic function [28,29,30,31,32]. Theoretically, EDA manifests the time and amplitude of stimuli at the skin level generated from control centers in the brain (hypothalamus) [33,34]. According to studies on humans [35], the frequency band limits of sympathetic nervous system activity are confined largely under the low-frequency ranges determined using power spectral density analysis of EDA signals. It was also recently proven that EDA responses to sympathetic stimulus were more consistent and reliable than the HRV indices in humans [36]. The possibility of developing an EDA-based objective biomarker for multi-level human pain assessment has been addressed and tested [37]. Moreover, a real-time pain detection algorithm was demonstrated using a wrist-worn EDA device and a smartphone [38]. EDA was also confirmed to be a good and reliable monitoring marker of acute stress levels [39]. More recently, EDA was investigated in rats [40] and humans [41] (under review) to demonstrate it as a physio-marker of early detection in central nervous system oxygen-toxicity-induced seizure application.

We hypothesize that EDA can be used as an objective tool for the assessment of stress in horses. In this study, we assess the feasibility of sympathetic arousal detection using EDA in horses.

## 2. Materials and Methods

### 2.1. Animals

For this study, 15 horses (6 geldings, 9 mares) of mean age 13.11 ± 5.4 years, mean body weight of 512.5 ± 60 kg in the University of Connecticut herd (9 Thoroughbreds, 4 Quarter Horses, and 2 Morgans) were studied. All horses were determined to be healthy based on physical examination. This study was reviewed and approved by the Institutional Animal Care and Use Committee (IACUC) of the University of Connecticut (protocol number A21-048) and complied with all the institutional, provincial, and national regulations on animal use in research.

### 2.2. Device Setup

Horses were outfitted with an own-designed elastic strap on their thorax that firmly held an EDA device (Shimmer GSR+, Shimmer Sensing, Dublin Ireland) using two hydrogel Silver/Silver Chloride (Ag/AgCl) electrodes placed just behind the left foreleg. The area under the electrodes was clipped and cleaned with alcohol. Figure 1 depicts the sensor placement on the animal. The signals were captured at a 128 Hz sampling frequency and Bluetooth-transmitted to a laptop to be stored and visualized in real-time.

### 2.3. Protocol

To investigate sympathetic arousal, EDA signals were recorded in response to two different tests: (1) Feeding test and (2) Startle test. The two tests were carried out in different sessions. The Feeding test provides a continuous (tonic) stimulus, while the Startle test is an acute stimulus. The idea is to show that EDA can serve as an indicator for long- and short-lasting sympathetic arousal. The device was connected 10 min before the data recording was started to allow animals to become accustomed to the device.

Feeding test: Feeding tests were carried out at the animal’s regular feeding time at 2:00 pm [42]. For each animal, a continuous 4 min segment was extracted, comprising 2 min of baseline (pre-feeding) data followed by a 2 min feeding stage. Animals were fed hay followed by grain on their normal feeding schedule in their assigned stall. Maintaining their normal schedule avoided additional disturbances that could create undesirable arousal linked to a change in their routine. The recording was performed for four animals at a time.Startle test: In the Startle test [22], the horses were taken one by one to a familiar, quiet, covered arena by an experienced handler familiar to the horses. The same handler handled all horses used. The horse was stood quietly in front of the solid wall that surrounds the arena. First, two minutes of baseline data were recorded. After that, a rainbow umbrella was opened abruptly from behind the wall and spun for one minute. The umbrella was positioned in the visual field of the horse. Data recording continued for two minutes after the conclusion of the spinning. Accordingly, the total length of the recording for the *Startle* test was 5 min (2 min baseline, 1 min after umbrella was opened, and 2 min after umbrella was removed).

### 2.4. EDA Signal Processing

#### 2.4.1. Preprocessing of EDA Data

The EDA signals were inspected to avoid including motion-artifact-induced responses visually and using the 3-axis accelerometer data. The EDA signals were down-sampled to 4 Hz and then filtered by a Butterworth lowpass filter of order five and a cutoff frequency of 1 Hz, followed by a median filter (1 s length) to remove high-frequency noise from the raw signal. The cutoff frequency was chosen to be conservative enough to capture the unseen sympathetic dynamics in the spectral analysis.

#### 2.4.2. Time-Domain Indices of EDA

Generally, EDA is analyzed in time domain identifying its tonic and phasic components [43,44]. The mean value of the tonic component is called the skin conductance level (SCL), and the rapid shifts observed in the phasic component are termed skin conductance responses (SCRs). We decomposed the EDA signal using the cvxEDA method [45]. The method is physiologically inspired and thoroughly explains EDA through a rigorous methodology based on Bayesian statistics, convex mathematical optimization, and sparsity. This method models the EDA signal as a summation of phasic and tonic components and an additive white Gaussian noise that incorporates model prediction error, measurement errors, and artifacts. The SCL and the number of non-specific SCRs (NS.SCRs) during a period of time are commonly used measures of sympathetic arousal based on EDA. The NS.SCRs were computed as the number of phasic drivers whose amplitude was greater than a threshold of 0.05 µs per minute.

#### 2.4.3. Spectral Analysis of EDA

Given that there is no prior information about the morphological and spectral distribution of EDA signal in horses, we first investigated these characteristics by comparing with our previously collected data in humans and rats [35,40]. For illustration purposes, samples of SCRs of humans [37] and horses (one of the subjects included in this study) are compared in Figure 2. Remarkably, the sample horse SCR was similar to the sample human SCR. The spectral content, in humans, was confined mainly below 0.15 Hz in resting conditions, 0.25 Hz under stress (postural, physical, and cognitive), and about 0.37 Hz under intense physical activity. For the rats, it was approximately twice as large as human EDA [35].

To further confirm our observations using spectral analysis, we evaluated EDA for the entire feeding session to ensure that the observation is generalizable. We applied a sampling frequency of 4 Hz that is high enough to capture the dynamic of sympathetic arousal. The power spectral analysis was performed by applying Welch’s periodogram method with 50% overlapped data. A Blackman window (length of 256 points) was used for each segment, the fast Fourier transform was calculated for each windowed segment, and the power spectra of the segments were averaged. Each spectrum was normalized, divided by its total power. Finally, we evaluated the frequencies containing 95% of the normalized spectral power across the animals. The power was confined mainly under 0.14 Hz. Figure 3 represents the ensemble plots of power spectrum distribution across the subjects.

#### 2.4.4. Statistical Analysis

Since feeding is a continuing or tonic stimulus, the long-lasting sympathetic arousal was quantified using the SCL and NS.SCRs. The SCL and NS.SCR were computed from each animal before feeding (pre-feeding) and during feeding. The differences in the obtained indices, before feeding and during feeding, were statistically evaluated.

For its part, the Startle test involved instantaneous stimuli, allowing the observation of short-time reactions. From each horse, the tonic and phasic component were extracted from a 100 s window starting 40 s before the stimulus and ending 60 s after the stimulus was applied. The mean values of tonic and phasic components were obtained for five consecutive 20 s segments from each signal. The differences between those 20 s segments were statistically evaluated.

The normality of the values were tested using the one-sample Kolmogorov–Smirnov test [46,47,48]. For normally distributed indices, we used a parametric test (*t*-test, *p* < 0.05) to evaluate the significance of the differences. For non-normally distributed indices, we used a nonparametric test (two-sided Wilcoxon rank sum test, 0.05 level of significance).

## 3. Results

Figure 4 shows the EDA signal and its components for a 2 min pre-feeding interval followed by a 2 min feeding period for a given horse. The differences in NS.SCRs between pre-feeding and feeding are visually evident.

For the Feeding test, as the SCL and NS.SCRs indices were found non-normally distributed using the Kolmogorov–Smirnov test, we used a two-sided Wilcoxon rank sum test at a 0.05 level of significance to evaluate the differences between pre-feeding and feeding. Figure 5 illustrates the resulting SCL and NS.SCRs obtained for the Feeding test. We found no significant differences in SCL (*p*-value = 0.1049), while NS.SCRs were significantly elevated during feeding compared to pre-feeding (*p*-value = 0.0281).

Figure 6 shows the ensemble average of the phasic component of all horses aligned using the startle time (the exact time at which the umbrella was opened) in such a way that the values where all set to zero at this particular reference time. A clear elevation in the phasic component of EDA can be observed across horses. The mean values of tonic and phasic components were found to be non-normally distributed. Hence, a two-sided Wilcoxon rank sum test was applied to evaluate the *p*-value of each pair of intervals at a 0.05 level of significance. Subsequently, Bonferroni correction was performed. The results indicate that the mean of the phasic component significantly increased in the segment right after starting the stimulus and remained elevated until the end of the test (*p*-value = 0.0039). The mean tonic component did not show significant differences between the 20 s segments. Figure 7 shows box plots for mean phasic responses within each segment.

## 4. Discussion

We have tested the feasibility of EDA to capture the sympathetic arousal produced by a long-lasting stimulus (Feeding test) and a short-lasting stimulus (umbrella Startle test) in horses. We found that EDA was able to detect sympathetic arousal elicited by both stimuli, especially its phasic component. In the Feeding test, we observed that the horses were more aroused during the feeding stage compared to the baseline, which was reflected on the phasic index of EDA (NS.SCRs). This was not observed in the tonic index of EDA (SCL), likely due to the high variability in the tonic component [49]. On the other hand, the umbrella Startle test was evaluated with HRV analysis and found powerful enough to stimulate an acute sympathetic activation [22]. The responses to this acute stimulus were reflected on EDA as a significant elevation in the phasic component of EDA after the onset of stimuli, while the animals showed 1 s latency between the onset of the stimulus and the EDA phasic response. The mean value of the phasic component did not drop significantly within the second and third 20 s intervals after the stimulus. The phasic component of EDA is the most suitable target for the observation of short-lasting stimulus such as a startle or acute pain in humans [37,50].

Our results suggest that EDA can be used to assess sympathetic stimuli continuously and objectively in horses and to develop tools to assess their welfare. Many of the currently available assessments of pain, stress, and welfare are at least somewhat subjective, including facial expressions [51,52,53] and body behaviors [54] whose sensitivity and evaluation can be affected by short observation periods and a lack of experience of the evaluator [55] Importantly, inter-observer agreement for facial expressions was lower than agreement on behaviors [56]. Continuous, objective monitoring may be useful in horses recovering from medical care, adjusting to new environments, or under stressful conditions, especially when personnel time is limited. Continuous monitoring may also allow for the earlier identification of stress than other assessments, although this requires further research [50,57,58,59]. Moreover, in clinical settings, EDA can be used as a reliable, sympathetic marker to improve the prediction of changes in arterial pressure in anaesthetized horses instead of, or in addition to, parasympathetic tone activity assessment [60,61].

HRV is the most widely used tool to assess autonomic function in humans and horses. In humans, it has been widely proven that the association between LF (0.045–0.15 Hz) components of HRV and sympathetic arousal are also reflected in the low-frequency components of EDA (0.045–0.25 Hz) [35]. In horses, the LF spectral contents of HRV (0.01–0.07 Hz) are considered the most relevant frequency ranges to sympathetic activation [62,63]. Looking into the spectral content of the EDA signal, similar to our observations in humans, we found that most of the power of EDA in horses is contained in the LF band (below 0.14 Hz). However, further research exploring EDA in response to other stimulus is needed to confirm the spectral content of the signal in horses.

As for the analysis of the EDA signal, we found that signal decomposition represents a challenge, as the current decomposition algorithms (cvxEDA, sparsEDA, Ledalab, psPM, etc. [44]) were developed for human EDA. In this study we used the cvxEDA for decomposition. The results (Figure 4, bottom panel) show the estimation of phasic drivers during baseline and feeding for a given horse. Although the results are correlated with the increase in phasic events (SCRs and drivers), the number of drivers is overestimated (multiple extra phasic components are wrongfully identified). Similar results were obtained with the other tools for EDA decomposition. Hence, although the current techniques helped us assess the elevation in phasic activity, an EDA decomposition algorithm specially designed for horse EDA would be ideal. More data are necessary for this purpose.

In this study, we collected EDA behind the left foreleg of the horses using a human wearable sensor. The results are promising; however, the amplitude and quality of the signal can be potentially improved by testing different locations, similar to previous research in humans [64]. A comprehensive and systematic study would be required to optimize the measurement location on the animal’s body in terms of signal quality and motion artifact robustness. Then, further methods for dealing with motion artifacts could be produced. We have previously developed methods for motion artifact detection and removal in human EDA [65,66]. In addition, an EDA recording device specially designed for the spectral characteristics of horse EDA will be developed in future studies to improve the quality of the signal.

## 5. Conclusions

In this study, we evaluated the feasibility of sympathetic arousal detection using EDA in horses. In particular, we used a Feeding test to induce overall arousal and a Startle test to induce acute stress. We found that the indices of the phasic component of EDA were sensitive to both tests and can potentially detect long-term and acute stress. Our results suggest that EDA has the potential to capture sympathetic activation in horses and provide a non-invasive continuous monitoring of stress, pain, and discomfort.

## Figures and Tables

**Figure 1 animals-13-00229-f001:**
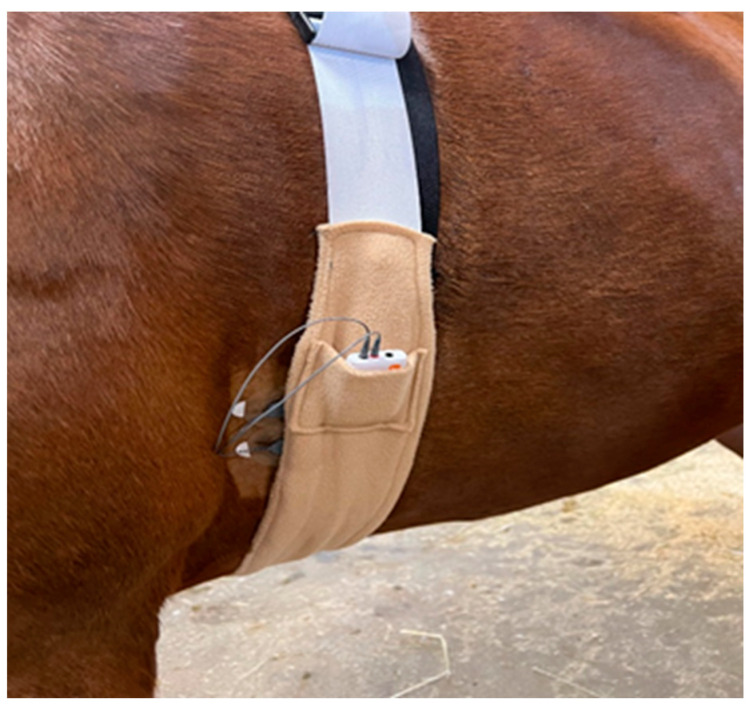
Device placement on the horse’s body.

**Figure 2 animals-13-00229-f002:**
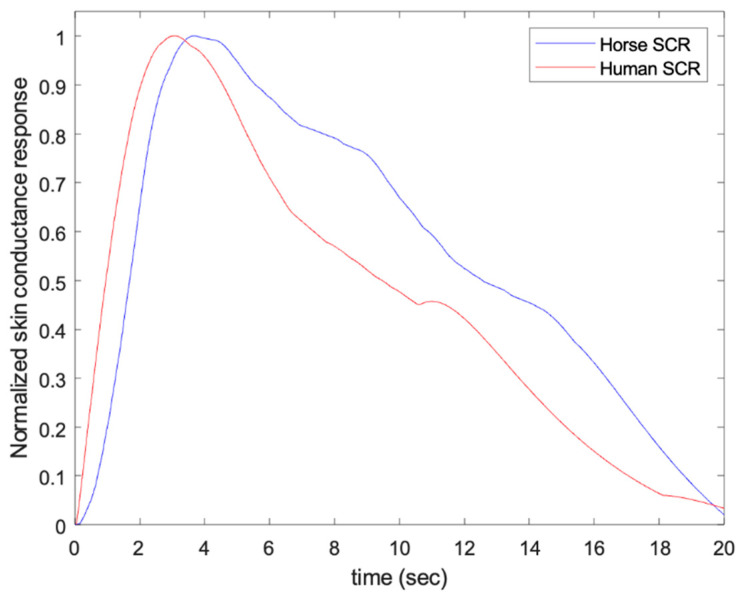
Samples of normalized skin conductance responses (SCRs) of human and horse.

**Figure 3 animals-13-00229-f003:**
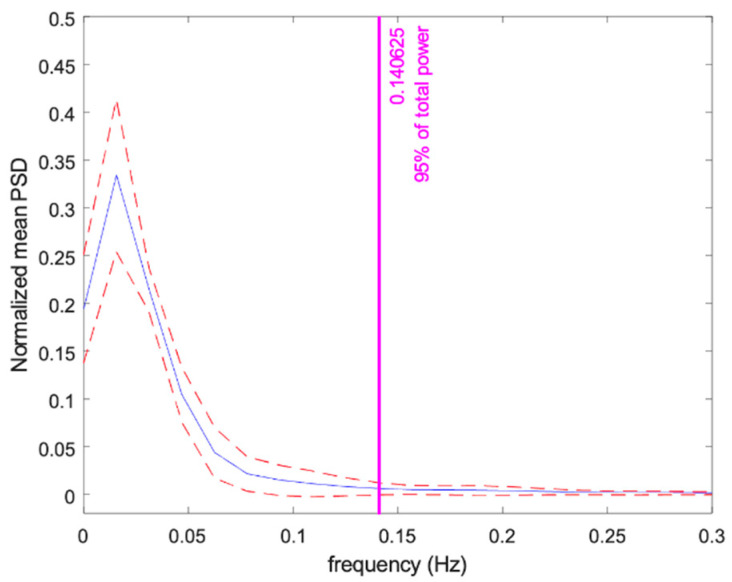
Ensemble of the normalized power spectrum distribution during feeding for N = 15 horses.

**Figure 4 animals-13-00229-f004:**
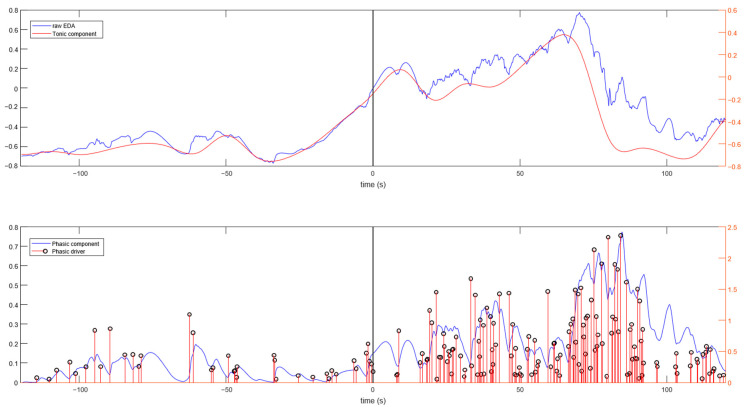
Raw EDA signal and tonic component (**top**), and phasic component and phasic drivers that are greater than 0.05 µS (**bottom**) before (baseline) and during the Feeding test for a sample subject.

**Figure 5 animals-13-00229-f005:**
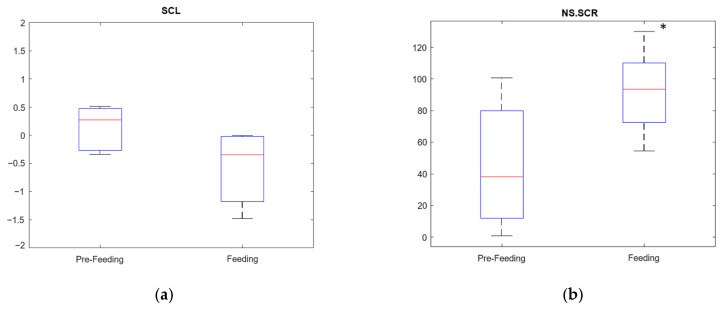
(**a**) Boxplot for SCL at pre-feeding and feeding stages (*p*-value = 0.1049). (**b**) Boxplot for NS.SCR at pre-feeding and feeding stages (*p*-value = 0.0281). * Represents statistically significantly differences (*p*-value < 0.05) between pre-feeding and feeding.

**Figure 6 animals-13-00229-f006:**
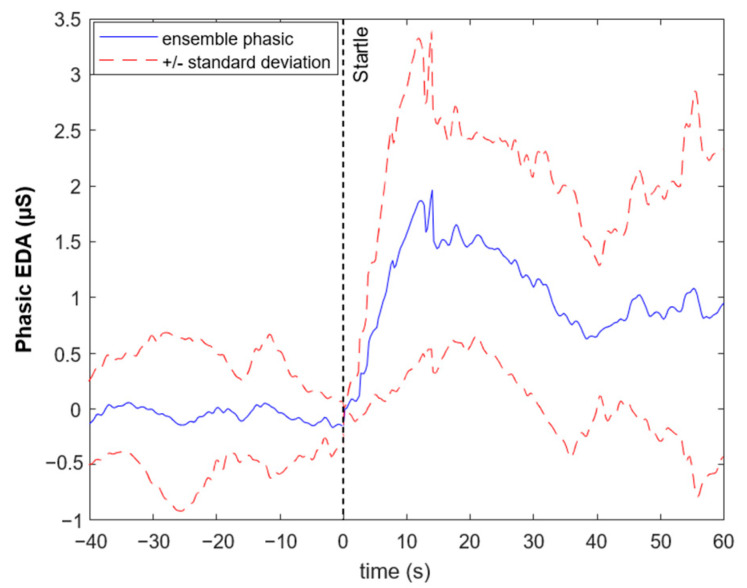
Ensemble of the phasic component for the Startle test from 40 s before the onset of the startle stimulus (umbrella) until 60 s after the stimulus.

**Figure 7 animals-13-00229-f007:**
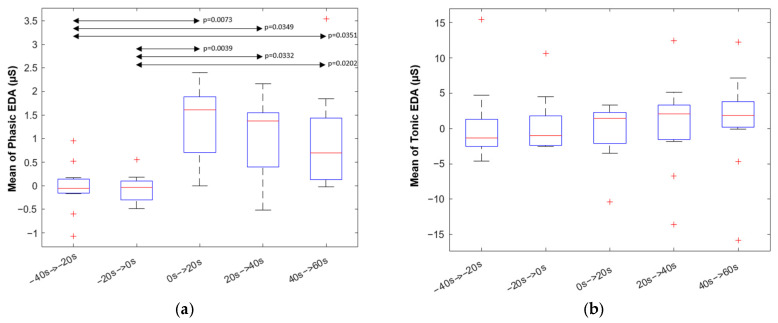
The boxplot of mean value of the phasic component of EDA (**a**) and tonic component of EDA (**b**) within 20 s segments starting from 40 s before the startle (umbrella) stimulus onset up to 60 s after stimulus onset. Significant differences (if *p* < 0.05) between segments are presented by a double arrow.

## Data Availability

The data presented in this study are available on request from the corresponding author.

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
