# Peer review of "Sympathetic Arousal Detection in Horses Using Electrodermal Activity"

_animals, 2023, doi:10.3390/ani13020229_

Round 1

Reviewer 1 Report

I have very much enjoyed reading this manuscript and am very excited to see the results. It makes me very hopeful that EDA could be a useful monitor for pain and other forms of distress for horses in the future. I have only a few minor comments.

Introduction

Line 67: Reference 23 is cited as relating to transport, but reference 23 relates to therapeutic riding sessions. I think this reference may be missing.

Line 69: Reference 24 should be reference 23.

Materials and Methods

Line 95: The different breeds:- 9 Thoroughbreds, 6 Quarter horses and 2 Morgan horses = 17 horses, but the total should be only 15 I think.

Line 103: ‘abdomen’ would be better as either ‘chest’ or ‘trunk’, as the photograph in Figure 1 shows the device around the ‘heart girth’ area.

Line 105: ‘left leg’ should be ‘left foreleg’ or ‘left thoracic limb’.

Line 121: ‘fed hay followed by grain’. How long after the hay was offered, was the grain introduced? That is, was the grain given immediately after hay was provided? Otherwise, if there was a delay between hay and grain provision, the horses might have been disturbed during their meal which might have caused a change in their arousal during the test?

Lines 128-129: ‘Data recording continued for 2 minutes after the conclusion of the spinning.’ I am a little confused. Does this mean that 5 minutes of recording was made for each horse:- 2 min before the umbrella was opened, then 1 minute from the opening of the umbrella to the end of spinning the umbrella, and finally 2 minutes after cessation of spinning the umbrella?

Line 155: ‘in humans and horses are compared in Figure 2.’ Are the horses referred to in this statement different from the 15 horses used in the present study? If so, please include the reference/s for where these other horse data come/s from.

Lines 179-180: ‘We statistically evaluated the difference in the obtained indices before feeding and during feeding.’, should be, ‘The differences in the obtained indices, before feeding and during feeding, were statistically evaluated.’.

Lines 185-186: ‘We statistically evaluated the difference between those 20-second segments.’ should be, ‘The difference between those 20-second segments was statistically evaluated.’.

Line 189: ‘nonnormally’ should be, ‘non-normally’.

Line 199: ‘non normally’, could be ‘non-normally’.

Results

Line 203: ‘elevated in during feeding’ should be, ‘elevated during feeding’.

Line 209: ‘the startle time’ – please could the authors expand this to explain that this (I think) means the 100 second epoch (40sec before to 60sec after the startle stimulus began).

Lines 213-214: ‘The results indicate…. significantly increases…’ should be, ‘The results indicated… significantly increased…’.

Figures

Figure 4: Please add to the figure legend that this data is from one horse. When the authors state ‘SCR’, is this the same as ‘NS.SCR’?

Figure 5: Do the authors have the P values for the significance?

Discussion

Line 272: ‘left leg’ should be, ‘left foreleg’ or ‘left thoracic limb’.

Conclusions

Line 288: ‘and provide with a non-invasive continuous monitoring of stress…’ should be, ‘and provide a non-invasive method for the continuous monitoring of stress…’

References

Reference 3 (Kaufman JM): the journal details are missing.

Please check if a reference is missing for the cited reference 23 – the paper should be about stress caused by restraint during transport.

Author Response

Dear reviewer,

We would like to express our gratitude to you for the favorable judgment of our work and for the comments, which helped to improve our paper. We include a point-by-point reply to each of the comments here. We also have modified our paper following these comments.  Please note that all changes to the text are highlighted in yellow.

Introduction

  • Line 67: Reference 23 is cited as relating to transport, but reference 23 relates to therapeutic riding sessions. I think this reference may be missing.

We thank the reviewer for the careful reading. The related reference is now cited properly. Consequently, reference 24 is corrected.

We have also reduced the statement accordingly:

“Other studies evaluated stress and temperamental traits in horses [5,23–25]. Physical effort was reported to affect HRV [25,26], reducing the specificity of the tool for stress assessment.”

  • Line 69: Reference 24 should be reference 23.

Thank you. We have addressed this in the first comment.

Materials and Methods

  • Line 95: The different breeds:- 9 Thoroughbreds, 6 Quarter horses and 2 Morgan horses = 17 horses, but the total should be only 15 I think.

We apologize for this typo. The correct breed breakdown is 9 thoroughbreds, 4 Quarter Horses, and 2 Morgans. We have corrected this in the manuscript.

  • Line 103: ‘abdomen’ would be better as either ‘chest’ or ‘trunk’, as the photograph in Figure 1 shows the device around the ‘heart girth’ area.

We thank the reviewer for the suggestion. The highlighted word was replaced in the manuscript as follows:  

‘’Horses were outfitted with an own-designed elastic strap on their thorax that firmly held an EDA device (Shimmer GSR+, Shimmer Sensing, Dublin Ireland) using two hydrogel Silver/Silver Chloride (Ag/AgCl) electrodes placed just behind the left foreleg.’’

  • Line 105: ‘left leg’ should be ‘left foreleg’ or ‘left thoracic limb’.

We thank the reviewer for the suggestion. It was responded in the 4th comment.

  • Line 121: ‘fed hay followed by grain’. How long after the hay was offered, was the grain introduced? That is, was the grain given immediately after hay was provided? Otherwise, if there was a delay between hay and grain provision, the horses might have been disturbed during their meal which might have caused a change in their arousal during the test?

Thanks for pointing out this. As we did not want to disturb the horses to minimize the unrelated stimulus effects, for this experiment we decided to follow their typical feeding regime in terms of order of food, time of the day, and delay between hay and grain. In other words, the delay between hay and grain provision was identical to what the animals had in their daily routine. The following text was added to the manuscript to explain this:

“Animals were fed hay followed by grain on their normal feeding schedule in their assigned stall. Maintaining their normal schedule avoided additional disturbances that could create undesirable  arousal linked to a change in their routine.”

  • Lines 128-129: ‘Data recording continued for 2 minutes after the conclusion of the spinning.’ I am a little confused. Does this mean that 5 minutes of recording was made for each horse:- 2 min before the umbrella was opened, then 1 minute from the opening of the umbrella to the end of spinning the umbrella, and finally 2 minutes after cessation of spinning the umbrella?

Thank you for your comment, and we are sorry for this confusion. As you mentioned, a total of 5 minutes of data was recorded for each horse in the Startle test, consisting of 2 minutes baseline, 1 minute from the opening of the umbrella, and 2 minutes after the cessation of spinning the umbrella. To make it clearer, we have modified the paragraph as follows:

“First, two minutes of baseline data were recorded. After that, a rainbow umbrella was opened abruptly from behind the wall and spun for one minute. The umbrella was positioned in the visual field of the horse. Data recording continued for two minutes after the conclusion of the spinning. Accordingly, the total length of the recording for the startle test was 5 minutes (2-min baseline, 1-min umbrella, 2-min after umbrella).”

  • Line 155: ‘in humans and horses are compared in Figure 2.’ Are the horses referred to in this statement different from the 15 horses used in the present study? If so, please include the reference/s for where these other horse data come/s from.

Thank you for this comment. This figure illustrates the similarities between a sample SCR from a horse that was collected in the present study and a human SCR from our previous publications.

The highlighted statement was added to the manuscript as below:

“For illustration purposes, samples of SCRs of humans  and horses (one of the subjects included in this study), were compared in Figure 2.” 

  • Lines 179-180: ‘We statistically evaluated the difference in the obtained indices before feeding and during feeding.’, should be, ‘The differences in the obtained indices, before feeding and during feeding, were statistically evaluated.’.

Thank you for this suggestion that helped us make the sentence clearer. We replaced the sentence in lines 179 and 180 accordingly

“The differences in the obtained indices, before feeding and during feeding, were statistically evaluated”

  • Lines 185-186: ‘We statistically evaluated the difference between those 20-second segments.’ should be, ‘The difference between those 20-second segments was statistically evaluated.’.

Thank you again for this suggestion. We replaced the sentence in lines 185 and 186 as shown below:

“The differences between those 20-second segments were statistically evaluated.”

  • Line 189: ‘nonnormally’ should be, ‘non-normally’.

Thanks for the correction. It was applied in the revised manuscript.

  • Line 199: ‘non normally’, could be ‘non-normally’.

Thanks for the correction. It was applied in the revised manuscript.

Results

  • Line 203: ‘elevated in during feeding’ should be, ‘elevated during feeding’.

Thanks for the correction. It was applied in the revised manuscript.

  • Line 209: ‘the startle time’ – please could the authors expand this to explain that this (I think) means the 100 second epoch (40sec before to 60sec after the startle stimulus began).

Thank you for noting this. The “Startle time” refers to the exact moment that the umbrella was opened abruptly. The phasic components were aligned according to this reference time in such a way that they were all shifted to zero level at this moment.

For clarification purposes, the highlighted sentence was added to line 209:

“Figure 6 shows the ensemble average of the phasic component of all horses aligned using the Startle time (the exact time in which the umbrella was opened) in such a way that the values were all set to zero at this particular reference time.”

  • Lines 213-214: ‘The results indicate…. significantly increases…’ should be, ‘The results indicated… significantly increased…’.

Thanks for this suggestion. The sentence was changed as highlighted below:

“The results indicated that the mean of the phasic component significantly increased in the segment right after starting the stimulus and remained elevated until the end of the test.”

Figures

  • Figure 4: Please add to the figure legend that this data is from one horse. When the authors state ‘SCR’, is this the same as ‘NS.SCR’?

Thanks for noting this information that needs clarification. The caption and legend in Figure 4 have been corrected for the accuracy in the terms as follows:

“Figure 4. Raw EDA signal and tonic component (top), phasic component and phasic drivers that are greater than 0.05 µs (bottom) before (baseline) and during the feeding test for a sample subject”

  • Figure 5: Do the authors have the P values for the significance?

Thanks for this comment. We have included the p-values for further clarification. The highlighted statements were added to the manuscript and figure 5 legend.

Lines 202-204:

“Figure 5 illustrates the resulting SCL and NS.SCRs obtained for the Feeding test. We found no significant differences in SCL (p-value = 0.1049), while NS.SCRs was significantly elevated in during feeding, compared to pre-feeding (p-value = 0.0281).”

Figure 5 legend:

“Figure 5. (a) boxplot for SCL at pre-feeding and feeding stages (p-value = 0.1049). (b) boxplot for NS.SCR at pre-feeding and feeding stages (p-value = 0.0281). * Represents statistically significant differences (p-value < 0.05) between pre-Feeding and Feeding.”

Discussion

  • Line 272: ‘left leg’ should be, ‘left foreleg’ or ‘left thoracic limb’.

Thanks for this comment. It was reworded as follows:

‘’In this study, we have collected the EDA behind the left foreleg of the horses using a human’s wearable sensor. ‘’  

Conclusions

  • Line 288: ‘and provide with a non-invasive continuous monitoring of stress…’ should be, ‘and provide a non-invasive method for the continuous monitoring of stress…’

Thanks for this suggestion. The sentence was changed as follows:

“Our results suggest that EDA has the potential to capture sympathetic activation in horses and provide a non-invasive continuous monitoring of stress, pain, and discomfort.”

References

  • Reference 3 (Kaufman JM): the journal details are missing.

Thanks for noting this. The full information of the “Master Thesis” was included in the revised manuscript.

  • Please check if a reference is missing for the cited reference 23 – the paper should be about stress caused by restraint during transport.

Thanks for noting this. We checked the references out to avoid any mistakes.

Reviewer 2 Report

Dear authors, 

Thank you for your paper. It was intructive and a great pleasure to review it. 

Author Response

Dear reviewer,

We would like to express our gratitude to you for your thorough review of our work and for the comments, which helped to improve our paper. We include a point-by-point reply to each of the comments here. We also have modified our paper following these comments. Hope that it is now suitable for publication. Please note that all changes to the text are highlighted in yellow.

Intro:

  • The introduction is maybe a little bit too long (e.g. I would remove all the applications of HRV [line 67-69]).

Thank you for your suggestion. We have reduced the Introduction section accordingly:

“Other studies evaluated stress and temperamental traits in horses [5,23–25]. Physical effort was reported to affect HRV [25,26], reducing the specificity of the tool for stress assessment.”

  • I would mention the other name of EDA, which is galvanic skin response.

We appreciate the reviewer for noting this. The highlighted statement was added to the revised manuscript to add GSR as an alternative name of EDA, as follows:

‘’HRV is also used in humans to assess autonomic arousal, but recently objective pain and stress monitoring has been implemented using electrodermal activity (EDA), as the EDA is a more sensitive and specific measure of sympathetic arousal than the HRV [7]. The EDA signal is also referred in the literature as galvanic skin response (GSR) and represents the conductance of the skin, which is modulated by sweat secretion (electrolyte solution) and has been gaining attention recently as a sensitive marker of sympathetic arousal, because sweat glands are innervated solely by the sympathetic nervous system.’’

  • Line 50: Could you please add that the conductance of the skin is varying due to sweat secretions (electrolyte solution) in order to help the reader to make the connections between EDA, sweat glands and sympathetic arousal?

Thanks for your comment. It has been addressed in response to the previous comment.

  • Line 64: « in [2] ». I would rather place this reference at the end of the sentence (line 66) as follows: “in frequency indices of HRV were found [2]”.

Thanks for this suggestion. It was applied in the revised manuscript.

  • Line 73: Sweat glands are innervated by the sympathetic branch of the ANS, but the EDA is not innervated in itself, even if the EDA is solely dependent on the sympathetic branch of the ANS. Could you please reword it?

Thanks for pointing this out. The statement was reworded as follows:

“For its part, the EDA is solely dependent on the sympathetic branch of the ANS because sweat glands are only innervated by this branch, which has made the EDA an ideal target for detecting emotions [27].”

  • Line 76-77: Are you sure that that stimuli are generated by control centers in the brain? Do you mean hypothalamus or brain stem? The sympathetic nervous system forms the thoraco-lumbar chain. I would suggest that you have a look at this review paper published in BJA Education in 2016 (Autonomic nervous system: anatomy, physiology, and relevance in anaesthesia and critical care medicine) or at the FRCA webpage (Anaesthesia UK : Autonomic Nervous System (frca.co.uk)).

Thank you for noting this. To avoid any ambiguity and making the sentence clearer, we reworded the statement as follows:

‘’ Theoretically, EDA manifests the time and amplitude of stimuli at the skin level generated from control centers in the brain (hypothalamus) [33,34].’’

  • Line 77-79: I guess that you refer to HRV when you mention power spectral density analysis (because you mention both sympathetic and parasympathetic components of the ANS, and not only the sympathetic component evaluated by EDA). If so, could you please make it clearer for the reader?

We apologize for this confusion. In the sentence you are referring to , the PSD analysis refers to EDA signals. Since the EDA is only affected by sympathetic nervous system, the SNS’s dynamics were determined by PSD in [34]. To clarify the ambiguity, the highlighted statement was added to the manuscript.

‘’ According to studies on humans [34], the frequency band limits of the sympathetic nervous system activity are confined largely under the low-frequency ranges determined using power spectral density analysis of EDA signals. ‘’

M & M:

  • Is the device validated? Or is it under development?

The Shimmer GSR+ is a validated and commercially available device.

  • Line 94: Could you please add the mean ± SD weight of the horses included in the study?

Mean weight ± SD for all 15 horses: 512.5 ± 60 kg; this has been added.

  • Line 96: I would not focus on the cardiovascular system but I would rather state that “All horses were determined to be healthy based on physical examination”.

Thanks for this suggestion. We have modified the sentence accordingly:

“All horses were determined to be healthy based on physical examination”

  • Line 100-101: Description of the tests performed on horses does not belong to Animals Could you please move it to a more appropriate section of the manuscript?

Lines 100 – 101 were removed from the Animals section.

  • Line 103: The elastic strap and the EDA device are placed just behind the elbow (Figure 1). Therefore, they are around the thorax and not around the abdomen.

‘’Horses were outfitted with an own-designed elastic strap on their thorax that firmly held an EDA device (Shimmer GSR+, Shimmer Sensing, Dublin Ireland) using two hydrogel Silver/Silver Chloride (Ag/AgCl) electrodes placed just behind the left foreleg.’’

  • Line 105: “Ag/AgCl”: abbreviations should be fully written when they first appear in the manuscript (unless otherwise recommended in the author guidelines). In addition, could you please add “just behind the left front leg” (same comment Line 272)?

Thanks for this comment. The abbreviations were fully written when they first appeared in the manuscript as follows:

Line 105: ‘’ … Silver/Silver Chloride (Ag/AgCl) …’’

Line 272: ‘’ … just behind the left foreleg …”

  • Line 118: An important information is missing: Was the protocol conducted at the place where horses were usually housed? If not, how much time were they were they given to get accustomed to the new place?

We thank the reviewer for noting this. The recordings were carried out at the same place that all the animals were housed. To make it clear, the highlighted statement was added as follows:

“Animals were fed hay followed by grain on their normal feeding schedule in their assigned stall. Maintaining their normal schedule avoided additional disturbances that could create undesirable  arousal linked to a change in their routine.”

  • Line 118: You stated that the recording was performed for four animals at a time. However, is it possible that non-fed horses may have seen or heard that horses were fed while they were not? If so, this may have changed the baseline level of sympathetic activity in non-fed horses.

We appreciate the reviewer for pointing this out. As we have explained in our response to your previous comment, we followed the same feeding schedule and location than the horses had daily, to avoid any disturbance. Is this also accounted for in the baseline?

  • Line 137: On which reference did you rely on to set the frequency of the filter?

Thank you for this question. Given that there is no prior information about the power distribution of EDA signals in horses, we chose a cutoff frequency of the low pass filter high enough to capture the possible high-frequency dynamics (1Hz), while still removing noise and other interferences. In our subsequent PSD analysis « Spectral Analysis of EDA », we found that 95% of the power is concentrated below 0.14 Hz, which is way below the cutoff frequency of the filter (1 Hz).

  • Line 139: Because you say that “generally, the EDA is analyzed”, could you please add references to support that?

We appreciate the reviewer for noting this. The references were cited in the revised manuscript.

  • Line 183: Could you please add a dash between 100 and second: “100-second window”?

Thanks for this comment. It was applied in the revised manuscript.

Results:

  • Line 197: Could you please change µs to µS (capital) in order to be consistent all along the manuscript?

Thanks for this comment. It was applied in the revised manuscript.

  • Line 203: Could you please add the p value for statistically significant results (unless otherwise recommended in author’s guidelines)? In addition, could you please remove the in: “elevated in during feeding”?

We thank you for this comment. We have added the p-value information as follows:

Lines 202-204:

“Figure 5 illustrates the resulting SCL and NS.SCRs obtained for the Feeding test. We found no significant differences in SCL (p-value = 0.1049), while NS.SCRs was significantly elevated in during feeding, compared to pre-feeding (p-value = 0.0281).”

  • Line 206: Could you please explain the meaning of * in the figure caption (i.e. p<0.05)?

We thank you for this comment. It was already applied in the revised manuscript as follows:

Figure 5 legend:

“Figure 5. (a) boxplot for SCL at pre-feeding and feeding stages (p-value = 0.1049). (b) boxplot for NS.SCR at pre-feeding and feeding stages (p-value = 0.0281). * Represents statistically significant differences (p-value < 0.05) between pre-Feeding and Feeding.”

  • Line 214: Could you please add the p value for statistically significant results (unless otherwise recommended in author’s guidelines)?

We appreciate the reviewer for the comment. The required information was included in the revised manuscript as follows:

“The results indicate that the mean of the phasic component significantly increases in the segment right after starting the stimulus and remained elevated until the end of the test (p-value = 0.0039.”

  • Line 215-216: “The mean tonic component did not show significant differences between the 20-second segments”. I would add a figure to support this statement, as you did for the tonic response in the feeding test.

Thanks for your suggestion. We have added a box plot of the mean of tonic EDA as shown in new Figure 7 (please see the revised manuscript). Consequently, the corresponding figure caption has been changed as follows:

“Figure 7. The boxplot of mean value of the phasic component of EDA (left) and tonic component of EDA (right) within 20-sec segments starting from 40 sec before the Startle (umbrella) stimulus onset up to 60-sec after stimulus onset. Significant differences (if p<0.05) between segments are presented by a double arrow. “

  • Line 219-220: The abbreviation of second is s and not sec. Could you please change it? Same comment for Line 223-224, Line 236.

Thanks for this comment. It was applied in the revised manuscript.

  • Figure 7: Could you please indicate (e.g. with *) statistically significant results?

Thanks for noting this. To apply the comment, the figure 7 was annotated by double arrows indicating significant differences (if p<0.05) as presented in response to the comment 23.

Discussion:

  • Line 228: I am wondering if it is correct to talk about sensitivity [true positive/(true positive+false negative)] from the statistical point of view (same comment Line 286). Indeed, you are not comparing your test with a gold standard test able to assess sympathetic arousal (even if I am not sure that there is a test considered as gold standard for assessment of sympathetic arousal)->microneurography (muscle sympathetic nerve activity MSNA) ( Hart EC, et al.. Recording sympathetic nerve activity in conscious humans and other mammals: guidelines and the road to standardization. Am J Physiol Heart Circ Physiol. 2017;312(5):H1031–H1051) (Macefield VG. Sympathetic microneurography. Handb Clin Neurol. 2013;117:353–364). Consequently, I would rather say that EDA is able to detect sympathetic arousal.

We thank you for this astute comment. We have modified line 228 accordingly:

“We found that the EDA was able to detect sympathetic arousal elicited by both stimuli, especially its phasic component.”

  • Line 249-251: “Continuous monitoring may also allow for earlier identification of stress than other assessments, although this requires further research”. Is this hypothesis supported by previous studies conducted in human medicine? If so, I would add the reference.

Thank you for this comment. We have added supporting references of studies conducted in humans:

    • Anusha, A. S., Jose Joy, S. P. Preejith, Jayaraj Joseph, and Mohanasankar Sivaprakasam. “Differential Effects of Physical and Psychological Stressors on Electrodermal Activity.” Conference Proceedings: ... Annual International Conference of the IEEE Engineering in Medicine and Biology Society. IEEE Engineering in Medicine and Biology Society. Annual Conference 2017 (2017): 4549–52. https://doi.org/10.1109/EMBC.2017.8037868.
    • Posada–Quintero, Hugo F., Youngsun Kong, and Ki H. Chon. “Objective Pain Stimulation Intensity and Pain Sensation Assessment Using Machine Learning Classification and Regression Based on Electrodermal Activity.” American Journal of Physiology-Regulatory, Integrative and Comparative Physiology 321, no. 2 (August 1, 2021): R186–96. https://doi.org/10.1152/ajpregu.00094.2021.
    • Al Machot, Fadi, Ali Elmachot, Mouhannad Ali, Elyan Al Machot, and Kyandoghere Kyamakya. “A Deep-Learning Model for Subject-Independent Human Emotion Recognition Using Electrodermal Activity Sensors.” Sensors (Basel, Switzerland) 19, no. 7 (April 7, 2019). https://doi.org/10.3390/s19071659.
    • Poh, Ming-Zher. “Continuous Assessment of Epileptic Seizures with Wrist-Worn Biosensors.” Thesis, Massachusetts Institute of Technology, 2011. http://dspace.mit.edu/handle/1721.1/68456.

  • In order to add further potential application of EDA in a clinical setting (especially in anaesthesia), I would recommend that you have a look at the following paper: Mansour C, Mocci R, Santangelo B, Sredensek J, Chaaya R, Allaouchiche B, Bonnet-Garin JM, Boselli E, Junot S. Performance of the Parasympathetic Tone Activity (PTA) index to predict changes in mean arterial pressure in anaesthetized horses with different health conditions. Res Vet Sci. 2021 Oct; 139:43-50. doi: 10.1016/j.rvsc.2021.07.005. Epub 2021 Jul 3. PMID: 34246942.
  • In order to compare EDA with PTA, and see potential advantages of EDA over PTA, I would recommend that you have a look at the following paper: Ruiz-Lopez P et al. Parasympathetic Tone Changes in Anesthetized Horses after Surgical Stimulation, and Morphine, Ketamine, and Dobutamine Administration. Animals 2022; 12 (8):1038.

In response to the previous two comments:

Thanks for these interesting suggestions! We are looking forward to exploring these exciting potential applications of the EDA in our future endeavors.  We have added the following statement regarding this interesting potential application of EDA.

Line 247: ‘’ Continuous, objective monitoring may be useful in horses recovering from medical care, adjusting to new environments, or under stressful conditions, especially when personnel time is limited. Continuous monitoring may also allow for earlier identification of stress than other assessments, although this requires further research [50,57–59].  Moreover, in clinical settings the EDA can be used as a reliable sympathetic marker to improve the prediction of changes in arterial pressure in anaesthetized horses, instead or in addition to parasympathetic tone activity assessment [60,61]”.